# Species Identification of Red Deer (*Cervus elaphus*), Roe Deer (*Capreolus capreolus*), and Water Deer (*Hydropotes inermis*) Using Capillary Electrophoresis-Based Multiplex PCR

**DOI:** 10.3390/foods9080982

**Published:** 2020-07-23

**Authors:** Mi-Ju Kim, Yu-Min Lee, Seung-Man Suh, Hae-Yeong Kim

**Affiliations:** Institute of Life Sciences & Resources and Department of Food Science & Biotechnology, Kyung Hee University, Yongin 17104, Korea; mijukim79@gmail.com (M.-J.K.); lym5373@naver.com (Y.-M.L.); teri2gogo@naver.com (S.-M.S.)

**Keywords:** red deer, roe deer, water deer, multiplex PCR, capillary electrophoresis

## Abstract

To provide consumers correct information on meat species, specific and sensitive detection methods are needed. Thus, we developed a capillary electrophoresis-based multiplex PCR assay to simultaneously detect red deer (*Cervus elaphus*), roe deer (*Capreolus capreolus*), and water deer (*Hydropotes inermis*). Specific primer sets for these three species were newly designed. Each primer set only amplified target species without any reactivity against non-target species. To identify multiple targets in a single reaction, multiplex PCR was optimized and combined with capillary electrophoresis to increase resolution and accuracy for the detection of multiple targets. The detection levels of this assay were 0.1 pg for red deer and roe deer and 1 pg for water deer. In addition, its applicability was demonstrated using various concentrations of meat DNA mixtures. Consequently, as low as 0.1% of the target species was detectable using the developed method. This capillary electrophoresis-based multiplex PCR assay for simultaneous detection of three types of deer meat could authenticate deer species labeled on products, thus protecting consumers from meat adulteration.

## 1. Introduction

The inaccurate information on the meat species in meat products has been globally concerned by consumers and regulatory agencies [1,2]. Since it is illegal to substitute meat species undeclared on the label of meat products, food manufactures must authenticate correct ingredients declared on their products [3,4]. In the meat industry, game meat consumed commercially is more expensive than meat from domesticated animals. This is because game meat has high nutritional value, such as higher protein and lower fat levels. In addition, it does not contain residues of antibiotics or growth hormones [3,5,6]. Accordingly, replacing game meat with relatively cheaper domesticated meat has taken place for the economic benefit [5]. For game meat products containing deer species, red deer (*Cervus elaphus*) and roe deer (*Capreolus capreolus*) are commonly used, meaning that these species are particularly susceptible to fraudulent labeling [7,8]. Several European countries traditionally permit game hunting [7]. Meanwhile, in Korea, wild animals, such as water deer (*Hydropotes inermis*), that damage crops can be temporarily hunted. However, their distribution and sale are limited, according to the Ministry of Environment guideline. In addition, water deer cannot be used as raw meat or processed food in Korea. To prevent food adulteration, an authentication method for differentiating red deer, roe deer, and water deer is essential.

Methods for detecting meat species have been developed based on DNA molecules and proteins [1,9]. Protein-based methods for deer species authentication have been used by enzyme-linked immunosorbent assay (ELISA), high-performance liquid chromatography (HPLC), and liquid chromatography-mass spectrometry (LC-MS) [10,11,12]. However, the thermal stability of nucleic acids compared to proteins can increase the amplification efficiency of target species in processed foods [13,14]. PCR, a representative DNA-based detection method, has been utilized for species identification in various fields [15,16,17,18]. For deer species, PCR-based detection methods, such as conventional PCR and real-time PCR, have been developed [3,8,19]. To differentiate closely related animal species, the development of specific primers for a target species is very crucial. Mitochondrial DNAs, such as cytochrome b, 12 S rRNA, and D-loop, are commonly used as target genes due to their sequence variations [2,20,21,22]. Furthermore, to increase the sensitivity of the DNA-based detection method in processed foods, a short fragment of PCR amplification is required because of DNA degradation during the manufacturing process [22,23]. Meanwhile, a multiplex PCR can simultaneously detect several species in a single reaction tube, resulting in effective detection [15,24,25]. Recently, to clearly separate similar sizes of amplicons of short PCR products, multiplex PCR methods combined with capillary electrophoresis have been developed and applied to simultaneously identify various target species [15,26].

The aim of this study was to develop a capillary electrophoresis-based multiplex PCR (CE-mPCR) method to verify the presence of wild animal species, such as red deer, roe deer, and water deer, in processed foods. The developed assay not only saves time and labor because it can simultaneously detect three target species but also can be utilized as a specific and sensitive method for a clear separation of these three species.

## 2. Materials and Methods

### 2.1. Sample Preparation

Raw tissue samples of 10 animal species (red deer: *Cervus elaphus*, water deer: *Hydropotes inermis*, roe deer: *Capreolus capreolus*, beef: *Bos taurus*, pork: *Sus scrofa domestica*, lamb: *Ovis aries*, goat: *Capra hircus*, horse: *Equus caballus*, chicken: *Gallus gallus*, and duck: *Anas platyrhynchos*) were collected from the Conservation Genome Resource Bank (CGRB, Seoul, Korea) or purchased from online and local markets of Korea. All samples were cut into small pieces and immediately stored at −20 °C until analysis.

### 2.2. DNA Extraction

DNAs were extracted from meat samples of animal species and processed products using a DNeasy Blood and Tissue Kit (Qiagen, Hilden, Germany), according to the manufacturer’s instructions with slight modifications. For good quality of DNA, 25 mg of meat was ground, and all buffers for extraction were used at double quantity. The purity and concentration of extracted DNAs were measured with a Maestro spectrophotometer (Maestro, Las Vegas, NV, USA). DNAs with a 260/280 nm ratio between 1.8 and 2.0 were used as templates for PCR.

### 2.3. Primer Design

To select species-specific regions for red deer, roe deer, and water deer, nucleotide sequences of target genes of 19 various animals were downloaded from the GenBank database (Appendix A) and aligned using Clustal Omega program (http://www.ebi.ac.uk/Tools/msa/clustalo/) (Figure 1). Species-specific primer sets were newly designed using Primer Designer, version 3.0 (Scientific and Educational Software, Durham, NC, USA). Primers used in this study are listed in Table 1. They were synthesized by Bionics (Seoul, Korea).

### 2.4. Single and Multiplex PCR Conditions

Single PCR was performed in a 25 μL final volume containing 10 × Buffer (Bioneer, Daejeon, Korea), 10 mM of dNTPs (Bioneer), 5 units of Hot Start *Taq* DNA polymerase (Bioneer), 0.4 μM of each primer, and 10 ng of DNA template. PCR reaction was carried out in a thermal cycler (Model PC 808, ASTEC, Fukuoka, Japan) as follows: pre-denaturation at 95 °C for 5 min, followed by 35 cycles of 95 °C for 30 s, 60 °C for 30 s, and 72 °C for 30 s, with a final extension step at 72 °C for 5 min.

PCR mixture for multiplex PCR was similar to single PCR except that it used 10 units of Hot Start *Taq* DNA polymerase (Bioneer) and optimized concentrations of primers. Annealing temperature concentrations of primers were optimized, considering specificity between three deer species. The annealing temperatures were estimated at 58, 59, 60, and 61 °C, and the red deer/roe deer/water deer primers combinations were 0.2/0.4/0.4, 0.2/0.4/0.5, and 0.4/0.4/0.4 μM. Finally, 0.2 μM of primers for red deer and 0.4 μM of primers for roe deer and water deer were used for multiplex PCR. Multiplex PCR reactions were carried out under the same conditions as single PCR. All PCR amplicons were electrophoresed on 3% agarose gels stained with ethidium bromide at 150 V for 25 min and confirmed by capillary electrophoresis using an Agilent 2100 Bioanalyzer (Agilent Technologies, Santa Clara, CA, USA) with DNA 1000 Lab Chip kit (Agilent Technologies). Briefly, 1 μL of PCR product and 5 μL of markers were loaded into each of the 12 wells and applied with a gel-dye mix in the chip, which was run in the bioanalyzer.

### 2.5. Specificity and Sensitivity of Multiplex PCR

The specificity of each primer set was performed using DNAs (10 ng each) isolated from 10 animal samples, including red deer, roe deer, and water deer. The specificity of the developed multiplex PCR was conducted using DNAs of the three target species to determine whether there was any cross-reactivity between closely related species.

The sensitivity of multiplex PCR was estimated using serially diluted DNAs (from 10 ng to 0.01 pg per reaction) of the three target species. Detection limits were tested using meat DNA mixtures. The ratio of DNA used in the mixture is shown in Table 2. This test was validated independently using different PCR instruments by different operators. All PCR reactions included a positive control (target DNA) and negative control (no-template).

## 3. Results and Discussion

### 3.1. The Specificity of Newly Designed Species-Specific Primers

In this study, the species-specific primer sets targeting mitochondrial genes of cytochrome b, 12 S rRNA, and D-loop for red deer, roe deer, and water, respectively, were newly designed. As shown in Figure 1, sequences of each target species were compared with two closely related species and 16 other animal species. Considering the intraspecific variation of target species, each primer was selected to have specific sequences of target species (Figure 1). Primer design is very important in the development of multiplex PCR because the primer has to selectively amplify the target in a single reaction containing several primer sets [12]. For multiplex PCR, the sizes of PCR products amplified by each primer set were different for the three target species (79, 126, and 160 bp for red deer, roe deer, and water deer, respectively, Table 1). Each set of species-specific primers amplified only the target species without showing cross-reactivity with nine other species (Figure 2), demonstrating high primer specificity for the target species.

### 3.2. Specificity and Sensitivity of Capillary Electrophoresis-Based Multiplex PCR

Using these newly designed primers for the identification of red deer, roe deer, and water deer, a multiplex PCR was first optimized by adjusting the concentration of each primer and annealing temperature of PCR condition. The specificity of this optimized assay was then evaluated using DNAs isolated from 10 animal species. As shown in Figure 3, each primer set for red deer, roe deer, and water deer in CE-mPCR specifically amplified target species, showing a high resolution between target species. These results indicated that these red deer-, roe deer-, and water deer-specific primers were sufficient to differentiate these closely related three species by multiplex PCR without causing any cross-amplification against non-target species.

The sensitivity of this CE-mPCR developed in this study was evaluated using DNA at different amounts ranging from 10 ng to 0.01 pg. The results are shown in Figure 4. In lane 6 of Figure 4, peaks of electropherogram were detected for red deer and water deer, but the peak of roe deer was not detected in lane 6 and shown in lane 5. Therefore, sensitivities for red deer, roe deer, and water deer were 0.1, 1, and 0.1 pg, respectively. Such high sensitivity of this assay might lead to accurate and reliable detection and differentiation of meat from three target deer species.

### 3.3. Application and Validation of Capillary Electrophoresis-Based Multiplex PCR Using Meat DNA Mixtures

To determine detection limits of CE-mPCR and confirm its applicability to a real sample, two sets of meat DNA mixtures were prepared as follows: (1) roe deer and red deer commonly used as game meat were added in water deer to authenticate game meat species present in commercial deer meats, and (2) red deer was contaminated with roe deer and water deer to detect wild animal species not permitted commercially in several countries. As shown in Figure 5, the detection limit of this assay was at least 0.1% for roe deer and red deer in meat DNA mixtures. In another meat DNA mixture, as low as 0.1% of roe deer and water deer could be detected (Figure 6). Microchip-based capillary electrophoresis technology used in this study is known to provide better accuracy and resolution in multiple target detection [15]. In the present study, at a low concentration of 0.1% for water deer (lane 6 in Figure 6), the result obtained by capillary electrophoresis was clearer than a PCR band visualized on agarose gel (Appendix A). This can help overcome the drawback, such as a false-negative result. In addition, the detection limit was validated independently in duplicate. All results obtained through intra-laboratory validation analysis were similar. The 0.1% of roe deer and red deer mixed in water deer and 0.1% of roe deer and water deer mixed in red deer were detected in two independent PCR reactions using the developed primer sets. Thus, this CE-mPCR assay developed in this study was able to simultaneously detect up to 0.1% of red deer, roe deer, and water deer in meat DNA mixtures. Compared to the limit of detection of 0.1% for roe deer and red deer [4] and 0.5% for red deer [3], our method showed higher or similar sensitivity. Meanwhile, since this was the first study to apply a detection method for water deer to meat DNA mixtures, the detection limit of 0.1% of our developed method could not be compared to previous reports. However, this method might be sufficient to be utilized as a specific and sensitive molecular tool for monitoring these three types of deer meat.

## 4. Conclusions

The CE-mPCR assay developed in this study could successfully detect three types of deer meat. Its applicability for authentication of meat species was verified using various ratios of meat DNA mixtures. This method is simple and user-friendly. It has high specificity and sensitivity for the simultaneous detection of red deer, roe deer, and water deer. However, despite several advantages of this method developed, since it is utilized for only qualitative detection, further study is required to the application of real-time PCR to quantify meat of target deer species in processed game meat.

## Figures and Tables

**Figure 1 foods-09-00982-f001:**
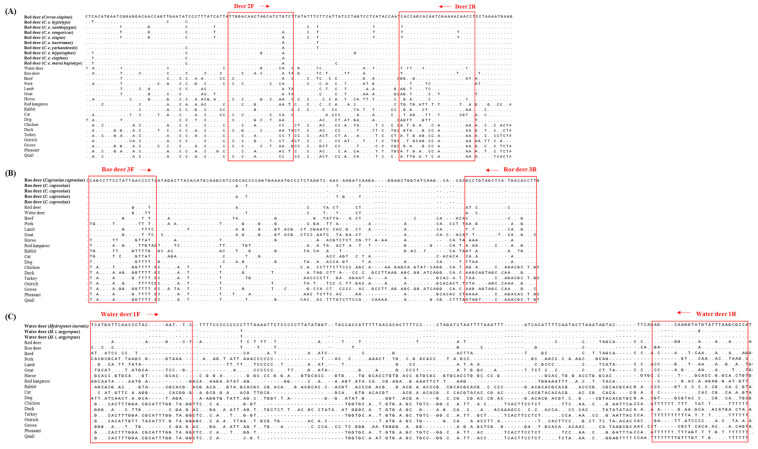
The sequence alignment of red deer (**A**), roe deer (**B**), and water deer (**C**) specific primers in the mitochondrial cytochrome b, 12 S rRNA, and D-loop regions against various animal species.

**Figure 2 foods-09-00982-f002:**
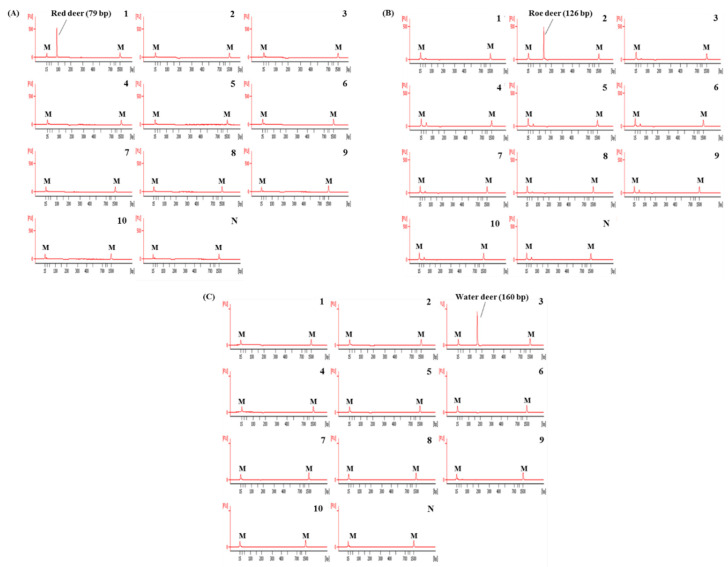
Electropherograms of specificity results of the single PCRs using newly designed primer sets for red deer (**A**), roe deer (**B**), and water deer (**C**). FU: fluorescence, M: alignment marker, lane 1: red deer, 2: roe deer, 3: water deer, 4: beef, 5: pork, 6: lamb, 7: goat, 8: horse, 9: chicken, 10: duck, and N: non-template.

**Figure 3 foods-09-00982-f003:**
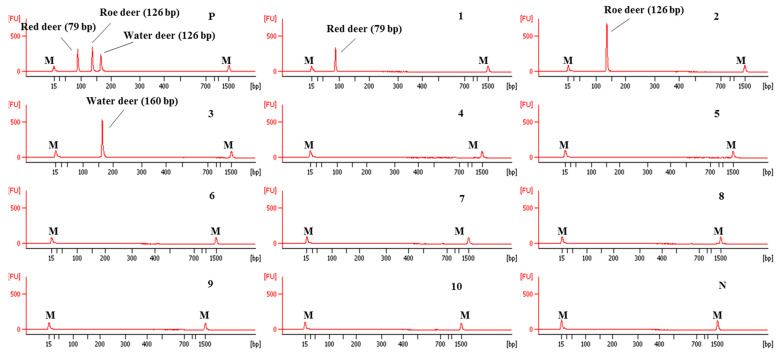
Electropherograms of specificity result of the multiplex PCR assay for red deer, roe deer, and water deer. FU: fluorescence, M: alignment marker, lane P: positive control (red deer, roe deer, and water deer), lane 1: red deer, 2: roe deer, 3: water deer, 4: beef, 5: pork, 6: lamb, 7: goat, 8: horse, 9: chicken, 10: duck, and N: non-template.

**Figure 4 foods-09-00982-f004:**
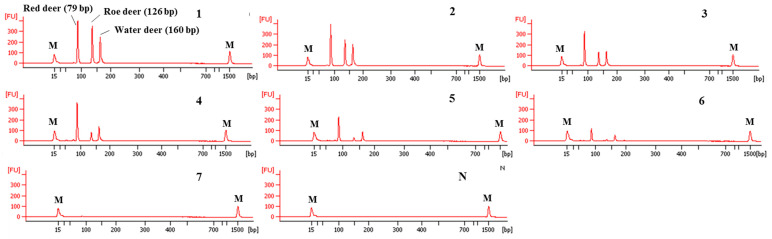
Electropherograms of sensitivity results of the multiplex PCR assay. FU: fluorescence, M: alignment marker, lanes 1–7: 1.0 × 10^1^, 10^0^, 10^−1^, 10^−2^, 10^−3^, 10^−4^, and 10^−5^ ng of three target species, and lane N: non-template.

**Figure 5 foods-09-00982-f005:**
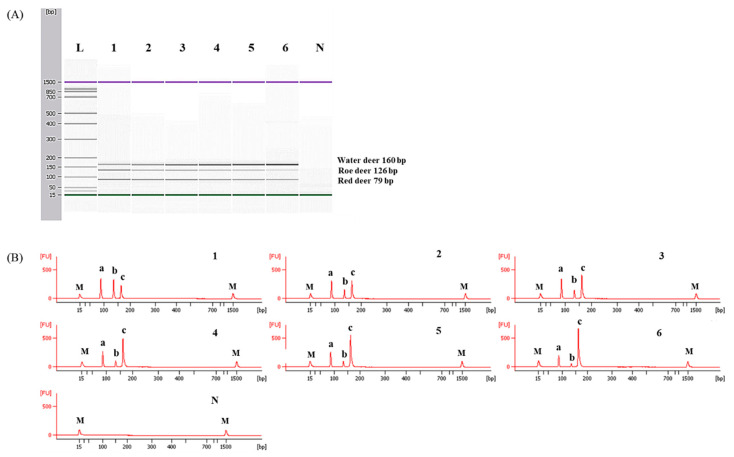
Detection limits of red deer and roe deer in water deer by the multiplex PCR assay. Gel image (**A**) and electropherograms (**B**). FU: fluorescence, M: alignment marker, lane L: 100 bp DNA ladder, lane 1: positive control (10 ng of DNA from target species), lanes 2–6: 10, 5, 1, 0.5, and 0.1% red deer and roe deer in water deer, and lane N: non-template. a, b, and c indicate red deer, roe deer, and water deer, respectively.

**Figure 6 foods-09-00982-f006:**
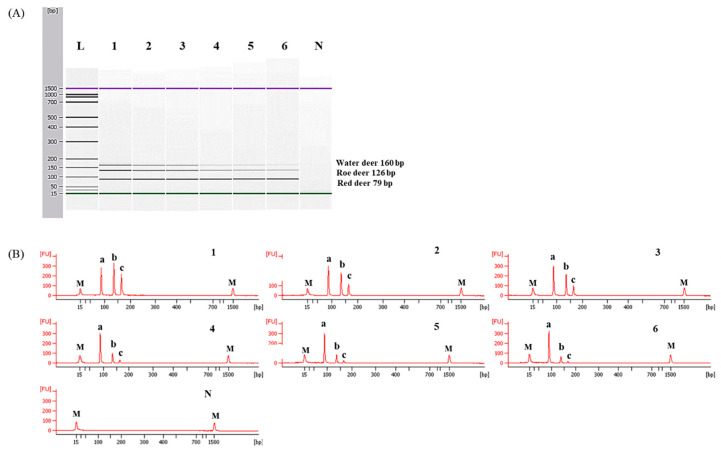
Detection limits of roe deer and water deer in red deer by the multiplex PCR assay. Gel image (**A**) and electropherograms (**B**). FU: fluorescence, M: alignment marker, lane L: 100 bp DNA ladder, lane 1: positive control (10 ng of each DNA from target species), lanes 2–6: 10, 5, 1, 0.5, and 0.1% roe deer and water deer in red deer, and lane N: non-template. a, b, and c indicate red deer, roe deer, and water deer, respectively.

**Table 1 foods-09-00982-t001:** Primers used in this study.

Target Species	Primer Name	Sequences (5′ → 3′)	Target Genes	Amplicon Size (bp)	Accession No.
Red deer	Deer 2 F	TGGACAACTAGCATCTGTCT	cyt b	79	JF489133.1
	Deer 2 R	GAGGTTGTTTTCGATTGTGCTGGTG			
Roe deer	Roe deer 3 F	CAGCCTTCCTATTAACCCCT	12 S rRNA	126	KJ681490.1
	Roe deer 3 R	AGGTGTCATGAGCTACAGGC			
Water deer	Water deer 1 F	CATGATTCAACCCTACAATTC	D-loop	160	NC011821
	Water deer 1 R	GGCGCTTAAATACATACCTTGCT			

**Table 2 foods-09-00982-t002:** The ratio of meat DNA mixtures used in this study.

	The Ratio of Meat DNA Mixtures (%)		
	Red Deer	Roe Deer	Water Deer
Roe deer and water deer in red deer	80	10	10
	90	5	5
	98	1	1
	99	0.5	0.5
	99.8	0.1	0.1
Red deer and roe deer in water deer	10	10	80
	5	5	90
	1	1	98
	0.5	0.5	99
	0.1	0.1	99.8

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
