# Peer review of "Species Identification of Red Deer (Cervus elaphus), Roe Deer (Capreolus capreolus), and Water Deer (Hydropotes inermis) Using Capillary Electrophoresis-Based Multiplex PCR"

_foods, 2020, doi:10.3390/foods9080982_

Round 1

Reviewer 1 Report

The present paper consists in an original research paper describing the development of a method for deer species authentication based on the sequences of animal barcode genes. Authors used sequences of the mitochondrial cytochrome b, 12S rRNA, and D-loop regions of red deer, roe deer, and water deer to design specific primers for each of the species, respectively. The methodology is appropriate and the research original however I find two main weaknesses in the work: 1\ The intraspecific variation was not considered since only one sequence of each species was used. This variation is common in animals at levels below the barcode species gap and should always be considered in the design of specific primers; and 2\ I think that to prove the applicability of the paper a market survey, or simply the application of the technique in a few real samples found in the market should be performed to validate the technique in realistic conditions.

A few recommendations to consider:

Introduction: I think that the reference of other method for deer species authentication in the introduction would improve the paper.

Lines 18: The two following sentences can be condensed in one. “The detection level of this assay was 0.1 pg for red deer and roe deer. It was 1 pg for water deer.”

Line 26: Please rephrase, it is not clear what authors mean and the English grammar needs revision. “Food adulteration about the description of the component in products has been globally concerned by consumers and regulatory agencies.”

Line 36: This idea is repeated: “Thus, game meat products are commercially available”.

Lines 62-63: Please format the species binomial names in italic.

Line 71: Please indicate the slight modifications introduced on the DNA extraction protocol.

Line 85: Please provide some details on the optimization of primers annealing temperature namely the results concerning specificity between the three deer species.

Line 90: Please correct the degree signal to º.

Figure 2: I do not agree with the use of a gel image representation as the technique used is electrophoresis and thus an electropherogram should be presented. I am aware that the equipment software does enable various representation however this are only artifacts and should be avoided. The presentation of electropherogram is enough and closer to results obtained. Authors stated in materials and methods that they run agarose gels of the amplicons thus they should be presented and not the electropherograms represented as agarose gels. I would be very happy to see agarose gel images as an indicator of the applicability of the method to the great majority of molecular biology labs with low resources. Additionally, in this image, indicate the molecular wight of the standards for all primers PCRs.

Figure 1: In the alignment C, for horse, there is some images superimpose that should be corrected.

Figure 3: I think there is a mistake in the indications of the MW of the standards. Again, only electropherograms should be presented.

Figure 4. The gel image representation is not necessary since the results obtained are in the form of electropherograms. The gel image is just a representation of a gel if samples were analyzed in eg. agarose, however, with very different sensitivities. The gel image representation is just an artifact and bring no improvement to the paper or interpretation. In my opinion only electropherogram should be presented. Also, the representation of species should not use letters as there is also an A and B image and that can be confusing.

Author Response

We are thankful to the referees for their deep and thorough review. Our detailed response to the review comments are given below. We have addressed all of the concerns of the reviewers and hope that the revised manuscript is now suitable for publication. 

Reviewer 1:

Comments and Suggestions for Authors

The present paper consists in an original research paper describing the development of a method for deer species authentication based on the sequences of animal barcode genes. Authors used sequences of the mitochondrial cytochrome b, 12S rRNA, and D-loop regions of red deer, roe deer, and water deer to design specific primers for each of the species, respectively. The methodology is appropriate and the research original however I find two main weaknesses in the work: 1\ The intraspecific variation was not considered since only one sequence of each species was used. This variation is common in animals at levels below the barcode species gap and should always be considered in the design of specific primers; and 2\ I think that to prove the applicability of the paper a market survey or simply the application of the technique in a few real samples found in the market should be performed to validate the technique in realistic conditions.

Response: Thank you for your comments.

Comment 1: To avoid the intraspecific variation in each species was used, we aligned sequences of closely related species and included the alignment results in Fig. 1. In addition, we added the related sentences in lines 121-124 as follows:

Lines 121-124: As shown in Fig. 1, sequences of each target species were compared with those of closely related species and 18 other animal species. Considering the intraspecific variation of target species, each primer was selected to have specific sequences of target species (Fig. 1).

Comment 2: Due to the limitations of the real samples found on the market, we applied this technique using a reference meat mixture (0.1, 0.5, 1,5, and 10%) in section 3.3.

A few recommendations to consider:

Introduction: I think that the reference of other method for deer species authentication in the introduction would improve the paper.

Response: As you recommended, we added the related sentence in the introduction section using references as follows:

Lines 42-44: Protein-based methods for deer species authentication have been used by enzyme-linked immunosorbent assay (ELISA), high performance liquid chromatography (HPLC), and liquid chromatography-mass spectrometry (LC-MS) [16-18]. However,

Lines 47-48: For deer species, PCR-based detection methods such as conventional PCR and real-time PCR have been developed [3,8,19].

Lines 18: The two following sentences can be condensed in one. “The detection level of this assay was 0.1 pg for red deer and roe deer. It was 1 pg for water deer.”

Response: As you recommended, we revised the sentence in lines 17-18 as follows:

Lines 17-18: The detection levels of this assay were 0.1 pg for red deer and roe deer, and 1 pg for water deer.

Line 26: Please rephrase, it is not clear what authors mean and the English grammar needs revision. “Food adulteration about the description of the component in products has been globally concerned by consumers and regulatory agencies.”

Response: As you recommended, we rephrased the sentence in lines 26-27 as follows:

Lines 26-27: The inaccurate information on the meat species in meat products has been globally concerned by consumers and regulatory agencies.

Line 36: This idea is repeated: “Thus, game meat products are commercially available”.

Response: As you recommended, we deleted this repeated sentence.

Lines 62-63: Please format the species binomial names in italic.

Response: Thank you for your comments. We corrected the species binomial names in italic in lines 65-67 as follows:

Lines 65-67: (red deer: Cervus elaphus, water deer: Hydropotes inermis, roe deer: Capreolus capreolus, beef: Bos taurus, pork: Sus scrofa domestica, lamb: Ovis aries, goat: Capra hircus, horse: Equus caballus, chicken: Gallus gallus, and duck: Anas platyrhynchos)

Line 71: Please indicate the slight modifications introduced on the DNA extraction protocol.

Response: As you recommended, we indicated the slight modification introduced on the DNA extraction protocol in lines 73-74 as follows:

Line 73-74: For good quality of DNA, 25 mg of sample was ground, and all buffers for extraction were used at double quantity.

Line 85: Please provide some details on the optimization of primers annealing temperature namely the results concerning specificity between the three deer species.

Response: As you recommended, we provided the detailed information on the optimization of multiplex PCR condition in lines 96-100 as follows:

Lines 96-100: Annealing temperature concentrations of primers were optimized considering specificity between three deer species. The annealing temperatures was estimated at 58, 59, 60, and 61°C, and the red deer/roe deer/water deer primers combinations were 0.2/0.4/0.4, 0.2/0.4/0.5, and 0.4/0.4/0.4 μM. Finally, 0.2 μM of red deer and 0.4 μM of roe deer and water deer was used for multiplex PCR.

Line 90: Please correct the degree signal to º.

Response: As you recommended, we corrected the degree signal in lines 93-94.

Figure 2: I do not agree with the use of a gel image representation as the technique used is electrophoresis and thus an electropherogram should be presented. I am aware that the equipment software does enable various representation however this are only artifacts and should be avoided. The presentation of electropherogram is enough and closer to results obtained. Authors stated in materials and methods that they run agarose gels of the amplicons thus they should be presented and not the electropherograms represented as agarose gels. I would be very happy to see agarose gel images as an indicator of the applicability of the method to the great majority of molecular biology labs with low resources. Additionally, in this image, indicate the molecular wight of the standards for all primers PCRs.

Response: As you recommended, we presented only electropherogram in Figure 2.

Figure 1: In the alignment C, for horse, there is some images superimpose that should be corrected.

Response: Thank you for your comment. We corrected Figure 1.

Figure 3: I think there is a mistake in the indications of the MW of the standards. Again, only electropherograms should be presented.

Response: As you recommended, we presented only electropherogram in Figure 3.

Figure 4. The gel image representation is not necessary since the results obtained are in the form of electropherograms. The gel image is just a representation of a gel if samples were analyzed in eg. agarose, however, with very different sensitivities. The gel image representation is just an artifact and bring no improvement to the paper or interpretation. In my opinion only electropherogram should be presented. Also, the representation of species should not use letters as there is also an A and B image and that can be confusing.

Response: As you recommended, we deleted gel image and presented only electropherogram in Figure 4. Also, we corrected the related sentence in lines 146-147 as follows:

Lines 146-147: In lane 7 of Fig. 4, peaks of electropherogram

Reviewer 2 Report

Attached file

Author Response

We are thankful to the referees for their deep and thorough review. Our detailed response to the review comments are given below. We have addressed all of the concerns of the reviewers and hope that the revised manuscript is now suitable for publication.

Reviewer 2:

General Comments

The method is a qualitative test and is useful to prove if water deer was used to produce any foods, since this game meat is forbidden to be processed for human consumption.

The authors have proved that the limit of detection of each game meat was down to 0.1% in meat DNA mixture. It sounds like the authors intended to use this method to detect food fraud of processing game meat in retail (such as sausage, ground meat etc., those the structure of meat was changed), which was actually the aim of this study. However, these types of food in retail were not included in the study.

Major:

  1. The methods were performed well under the control/known conditions. It would be good to find any foods or products in retail which legally contain any of these game meats to prove the applicability of the method with field samples, since those products may contain not only meat, but also other matrix, which may influence the results.

Response: Since many studies have been reported that DNA is stable under highly processed conditions such as high temperature and pressure, this DNA-based method developed in this study may be sufficient to detect the deer species in processing game meat in retail. And, in our previous reports, we examined the effect of processed conditions and/or food matrix such as food additives and seasoning, and there was few PCR efficiency change (Qin et al., 2015, Kim and Kim, 2017, Kim et al., 2018). Thus, in this study, we validated the applicability of this method through intra-laboratory analysis.

  1. The methods are quality tests, that means you cannot differentiate how many percent or Gram of target meat in meat products. If producer said, they used 20% of red deer/roe deer meat (legally used) but indeed only 2%, by using this method, it will be positive for red deer but it is still food fraud. Please mention about the limitation of the test in the manuscript and the suggestion for the next study such as the application of multiplex quantitative PCR.

Response: As you recommended, we mentioned about the limitation of the test and suggested the further study in lines 193-195 as follows:

Lines 193-195: However, despite several advantages of this method developed, since it is utilized for only qualitative detection, further study is required to the application of real-time PCR for measuring the quantity of deer species.

Minor:

Avoid using “meat mixture”, if you mix the DNA extract of each meat. Instead of that please using “meat DNA mixture”.

Response: As you recommended, we corrected “meat mixture” to “meat DNA mixture” in the entire manuscript.

Line 62: „elaphus“, not “elephus”

Response: Thank you for your comments. We corrected to “elaphus” in line 65.

Lines 62 – 64: Scientific name, please write it italic.

Response: Thank you for your comments. We corrected the scientific names in italic in lines 65-67 as follows:

Lines 65-67: (red deer: Cervus elaphus, water deer: Hydropotes inermis, roe deer: Capreolus capreolus, beef: Bos taurus, pork: Sus scrofa domestica, lamb: Ovis aries, goat: Capra hircus, horse: Equus caballus, chicken: Gallus gallus, and duck: Anas platyrhynchos)

Line 71: DNA extraction

  • Please explain more about “slight modification”
  • How many Gram meat used for DNA extraction? did you treat meat e.g. were they ground?

Response: As you recommended, we explained more about the DNA extraction protocol used in this study as follows:

Lines 73-74: For good quality of DNA, 25 mg of sample was ground, and all buffers for extraction were used at double quantity.

Line 80: “Korea”?

Response: Thank you for your comments. We corrected into “Korea” in line 83.

Line 84: Table 1

  • Delete column “Final conc. (µM)” and move the information to section 2.4, line 87
  • Delete column “Reference”
  • Primer for Red deer = 0.2 µM, nut in Text (Line 87) = 0.4 µM. Which one is right?

Response: As you recommended, we deleted the “Final con.” and “Reference” columns and the information on the optimized concentrations of primers was written in lines 99-100 as follows:

Lines 99-100: Finally, 0.2 μM of red deer and 0.4 μM of roe deer and water deer was used for multiplex PCR.

Line 94- 95: Please explain more about

  • Agarose gel electrophoresis (e.g. Agarose concentration, Volt and time of running)
  • Capillary electrophoresis

Response: As you recommended, we explained the detailed information on agarose gel and capillary electrophoresis in lines 101-105 as follows:

Lines 101-105: All PCR amplicons were electrophoresed on 3% agarose gels at 150 V for 25 min and confirmed by capillary electrophoresis using an Agilent 2100 Bioanalyzer (Agilent Technologies, Santa Clara, CA, USA) with DNA 1000 Lab Chip kit (Agilent Technologies). Briefly, 1 μL of PCR product and 5 μL of markers were loaded into each of the 12 wells and applied with gel-dye mix in the chip, which was run in the bioanalyzer.

Line 107 – 108: It is not clear what are “++” in the table. If they are the results from other operators and the intensity of the peak was not analyzed, this column should be deleted and results should be presented in Text within Section 3.3. Additionally, please mention the name of institute of other operators?

Response: As you recommended, we deleted the “result” column in Table 2 and presented the results in section 3.3 as follows:

Lines 170-173: In addition, the detection limit was validated independently in duplicate. All results obtained through intra-laboratory validation analysis were the same. 0.1% of roe deer and red deer mixed in water deer and 0.1% of roe deer and water deer mixed in red deer were detected in two independent PCR reactions.

Line 124:         

  • There is no Lane L, but Lane M
  • Please indicate that it is a Fig of Agarose gel (Figure 2: Agarose gel electrophoresis show specificity results…..)

Response: As you recommended, we corrected the error and indicated the detailed information in Figure 2.

Line 136:           

  • There is no Lane L, but Lane M
  • Lane P: positive control. Please describe what is this positive control?
  • Please indicate that it is a Fig of Agarose gel (Figure 2: Agarose gel electrophoresis show a specificity result….. )

Response: As you recommended, we corrected the error and indicated the detailed information in Figure 3.

Lane 139:

  • Do you mean Lane 6?

Response: As you recommended, we rewrote this sentence in lines 147-148 as follows:

Lines 147-148: but peak of roe deer was not detected in lane 7 and shown in lane 6.

Round 2

Reviewer 1 Report

na

Author Response

We would like to thank you for the time and effort taken to review our manuscript.

Reviewer 2 Report

There are some texts needed to be corrected 

Author Response

We are thankful to the referee for their deep and thorough review. Our detailed response to the review comments are given below. We have addressed all of the concerns of the reviewer and hope that the revised manuscript is now suitable for publication. 

Reviewer 2-2:

The authors corrected the manuscripts according to the recommendation after first review. However, there are some texts needed to be edited.

Minor:

Line 71:           please correct “from samples” to “from meat samples”

Response: As you recommended, we corrected to “from meat samples” in line 71 as follows:

Line 71: from meat samples

Line 73:          please correct “25 mg of sample” to “25 mg of meat”

Response: As you recommended, we corrected to “25 mg of meat” in line 73 as follows:

Line 73: 25 mg of meat

Line 99:            add “primers for” to “0.2 µM of primers for red deer…”

Response: As you recommended, we added “primers for” in line 99 as follows:

Line 99: 0.2 μM of primers for

Line 99:            add “primers for” to “0.4 µM of primers for roe deer….…”

Response: As you recommended, we added “primers for” in line 100 as follows:

Line 100: 0.4 μM of primers for

Line 122:       correct “with those of closely related species and 18 other ….” to “with two closely related deer species and 16 other ….”

Response: As you recommended, we corrected to “with two closely related deer species and 16 other” in line 122 as follows:

Line 122: with two closely related deer species and 16 other

Line 133:        correct the sentence as following: Electropherograms of specificity results of the single PCRs using newly designed primer sets for red deer (A), roe deer (B), and water deer (C).

Response: As you recommended, we corrected the sentence in line 133 as follows:

Line 133: Electropherograms of specificity results of the single PCRs using newly designed primer sets for red deer (A), roe deer (B), and water deer (C)

Line 146:       please correct “lane 7” to “lane 6” (because lane 6 = 10-4 ng = 0.1 pg; and lane 7 = 10-5 ng = 0.01 pg, this is according to your results in text, that the limit of detections were 0.1 and 1 ng)

Response: As you recommended, we corrected to “lane 6” in line 146 as follows:

Line 146: lane 6

Line 148:       please correct “lane 7” to “lane 6” and please correct “lane 6” to “lane 5” (see comment for line 146)

Response: As you recommended, we corrected to “lane 6” and “lane 5” in line 148 as follows:

Line 148: lane 6, lane 5

Line 149-150:   correct “…reliable detection of these three meat species.” to “..reliable detection and differentiation of meat from three target deer species.

Response: As you recommended, we corrected to “reliable detection and differentiation of meat from three target deer species” in lines 149-150 as follows:

Lines 149-150: reliable detection and differentiation of meat from three target deer species

Line 172:            please correct “the same” to “similar”

Response: As you recommended, we corrected to “similar” in line 172 as follows:

Line 172: similar

Line 173:            add these phrase “using the developed primer sets” after “PCR reactions…”

Response: As you recommended, we added the phrase in line 173 as follows:

Line 173: using the developed primer sets

Line 195:        delete the phase “…for measuring the quantity of deer species” and correct the phrase as following “…….of real-time PCR to quantify meat of target deer species in processed game meat.”

Response: As you recommended, we corrected the sentence in line 195 as follows:

Line 195: to quantify meat of target deer species in processed game meat